# A New Strategy for Disc Cutter Wear Status Perception Using Vibration Detection and Machine Learning

**DOI:** 10.3390/s22176686

**Published:** 2022-09-04

**Authors:** Xiaobo Pu, Lingxu Jia, Kedong Shang, Lei Chen, Tingting Yang, Liangwu Chen, Libin Gao, Linmao Qian

**Affiliations:** 1Tribology Research Institute, State Key Laboratory of Traction Power, Southwest Jiaotong University, Chengdu 610031, China; 2China Railway Engineering Equipment Group Technical Service Co., Ltd., Zhengzhou 450000, China

**Keywords:** wear, disc cutter, status monitoring, vibration, machine learning

## Abstract

Carrying out status monitoring and fault-diagnosis research on cutter-wear status is of great significance for real-time understanding of the health status of Tunnel Boring Machine (TBM) equipment and reducing downtime losses. In this work, we proposed a new method to diagnose the abnormal wear state of the disc cutter by using brain-like artificial intelligence to process and analyze the vibration signal in the dynamic contact between the disc cutter and the rock. This method is mainly aimed at realizing the diagnosis and identification of the abnormal wear state of the cutter, and is not aimed at the accurate measurement of the wear amount. The author believes that when the TBM is operating at full power, the cutting forces are very high and the rock is successively broken, resulting in a complex circumstance, which is inconvenient to vibration signal acquisition and transmission. If only a small thrust is applied, to make the cutters just contact with the rock (less penetration), then the cutters will run more smoothly and suffer less environmental interference, which would be beneficial to apply the method proposed in this paper to detect the state of the cutters. A specific example was to use the frequency-domain characteristics of the periodic vibration waveform during the contact between the cutter and the granite to identify the wear status (including normal wear state, wear failure state, angled wear failure state) of the disc cutter through the artificial neural network, and the diagnosis accuracy rate is 90%.

## 1. Introduction

Tunnel Boring Machine (TBM) is suitable for the one-time forming of the full section of long tunnels in complex geology, and plays a key role in the construction of engineering tunnels. The disc cutter is one core component of TBM, which is responsible for the rock-breaking function, and its state directly affects whether the TBM can operate normally [1]. Complicated and violent contact behavior between cutter and rock during the rock-breaking process causes the disc cutter to wear out easily. The form of wear can be divided into uniform wear and non-uniform wear. Non-uniform wear includes angled wear [2] and the broken edge of the cutter ring. Compared with uniform wear, non-uniform wear is more likely to cause serious damage to the cutter and further damage the cutter head. In the Nanjing Yangtze River Tunnel Project, the tool wear status was not recognized in time. The severely worn cutter caused damage to the cutter-head structure, and the repair work took half a year, which seriously affected the progress of the project [3]. In particular, the rock layer for TBM construction is relatively hard, and the cutter wear is greater, and the cutter cost accounts for a large proportion of the engineering cost. It is of great significance to pay attention to and strengthen cutter management. In the current actual construction project, the common method to identify and diagnose the wear status of the cutter is still to stop and open the warehouse for inspection. In an earth pressure balance tunnel excavation project in Seattle, USA, 40 fixed shutdown check points were set up during the construction of a 6.4 km tunnel, which incurred high economic costs [4]. Moreover, the working environment of high temperature and high pressure cannot effectively guarantee the health and safety of the workers in the warehouse. Therefore, there is an urgent need to establish a practical method for identifying, sensing and diagnosing the wear status of the disc cutter, so as to provide timely information on the wear status of the cutter.

At present, the existing methods for identifying and diagnosing the wear of TBM disc cutters are mainly based on the overall parameters of the cutter head, such as thrust, torque, speed, temperature rise, field penetration index, etc., to estimate the overall wear of the cutters, or to establish a lifetime-prediction model with the help of cutter ring properties, geological conditions and tunneling parameters [5,6,7,8,9,10,11,12,13,14,15,16,17]. For example, Rong et al. [5] established a cutter wear prediction model based on the two parameters of cutter head torque and foaming fluid flow. The original data came from 20 tunnel sections dominated by conglomerate and weathered rock formations in Chengdu subway construction. They used multiple regression analysis to establish the relationship between the two input parameters and the cutter wear status, and obtained an indirect prediction equation that can reasonably evaluate the overall wear of the cutters in gravel and weathered rock and the number of cutter changes per unit of excavation volume. These methods are used to make cutter consumption and reserve plans before construction, but are not suitable for real-time evaluation of the wear status of a single cutter.

In order to evaluate the wear status of a single cutter, some studies have developed different types of sensors, such as eddy-current sensors, ultrasonic sensors, laser sensors and device, and so on [18,19,20,21,22]. Lan et al. [18] proposed an online monitoring and identification system for cutter wear based on eddy-current sensors. By establishing the mapping relationship between eddy current distance and cutter wear, the cutter wear status can be monitored in real time, and the on-site real-time continuous test results show that the amount of wear error is less than 1 mm. Guo et al. [21] designed a stable cutter wear monitoring system based on the principle of ultrasonic thickness measurement. Zhang et al. [22] designed a cutter dynamic wear detection device based on the characteristic that the on-off of the parallel laser beam is regulated by the degree of wear of the cutter ring, using a laser transmitter and a photosensitive sensor. We have constructed Table 1 to sort out the advantages and disadvantages of the existing methods.

In the machinery-manufacturing industry, sensors that use force and vibration as measurement signals are rugged, cheap and easy to use, making them more suitable for industrial environments. According to statistics reported in the literature, the identification of cutter wear based on force and vibration signals has been the most widely used in the state monitoring of ordinary cutting tools [23]. Additionally, the use of force or vibration sensors can be flexibly loaded on cutter parts, clamping parts or cutter heads, and the frequency-domain and time-domain measurement ranges of the collected signals are wide, which is convenient for signal conversion processing to extract characteristic information, and has the advantage of being compatible with wireless communication technology. At present, researchers have realized the detection of the running state and fault characteristics of the cutter by collecting the vibration signal of the main bearing and the current signal of the main motor of the TBM cutter head [24]. This method effectively avoids the problems of signal distortion and mixed-signal background, and improves the accuracy of diagnosis. However, considering the complex vibration conditions that the disc cutter of TBM may face in rock breaking, there are still few studies on the detection and diagnosis of TBM disc cutter wear using the vibration sensor. Therefore, it has potential and application prospects to develop a new method for online monitoring of cutter wear status based on the vibration signal.

In recent years, artificial intelligence and algorithms have emerged and gradually shown sufficient potential and advantages in the field of intelligent fault diagnosis [25]. One of the typical representatives is the artificial neural network (ANN) [26,27,28,29,30,31,32,33]. In 1987, neural network technology was first combined with sensor technology to improve the reliability of tool monitoring [30]. Gao et al. [31] designed a method to monitor the wear status of milling cutters based on integrated neural networks. They used wavelet transform to process force and vibration signals, and through the fusion of signal combination and sub-network output decision-making, they meet the requirements of real-time monitoring of tool wear status. Zhou [32] established an experimental system for monitoring the wear status of turning tools using force sensors and vibration sensors. By collecting vibration signals and cutting-force signals related to tool wear status, time-domain and frequency-domain analysis are performed to extract sensitive features, and genetic algorithm is introduced into BP neural network, which effectively realizes the monitoring of turning tool wear status. Xie et al. [33] proposed a milling cutter wear status recognition and prediction model based on Stacking Sparse De-noising Auto-Encoder (SSDAE) technology and PSO-LSSVM model. The time domain and frequency domain features were extracted with a high correlation coefficient higher than 0.95, and good prediction accuracy was obtained. At present, the ANN algorithm has some applications in the field of disc cutter wear perception [34,35]. Yu et al. [35] established a one-dimensional convolutional neural network model for evaluating the wear of disc cutters based on field parameters, and verified the average accuracy of 87.8% using Mumbai subway field data. Therefore, it is feasible and reliable to use the artificial neural network algorithm model to diagnose and identify the wear of industrial tools.

Although both deep-learning algorithms and vibration-signal analysis are valuable in the wear diagnosis of conventional tools, there are few studies and applications in TBM disc cutter wear diagnosis. On the basis of previous research [24,25,26,27,28,29,30,31,32,33,34,35], this paper is the first to propose a diagnosis and identification method for the abnormal wear state of TBM single-edge disc cutter based on a deep-learning algorithm combined with a vibration signal analysis. With the help of the neural network model, the problem that it is difficult to accurately identify the vibration characteristics of the cutter under noise interference is solved, and the correlation between the complex vibration characteristics and the wear status of the cutter is effectively established.

In this work, the three-axis acceleration sensor installed on the cutter part and the clamping part was used to collect the vibration signal generated during the cutter breaking rock process, and the fast Fourier transform (FFT) algorithm was used to convert the vibration signal to time-frequency to obtain the frequency domain information. This work analyzed and extracted time-frequency domain features, established the correlation between vibration features and wear status, and introduced ANN to realize intelligent perception and recognition. The process of wear perception is shown in Figure 1. The overall logic of this research is shown in Appendix A. 

## 2. Material and Methods

### 2.1. Test Cutter and Rock Sample

The methodology adopts a 19 in. single-edged disc cutter. There are three wear states for the disc cutter in total. The three types of cutter are all disassembled from the TBM equipment in the actual construction project, as shown in Figure 2a. According to the current standard, the radial limit wear of 19 in. positive cutters is 35 mm (Figure 2b). In reference to normal wear of the cutter, after a certain degree of uniform wear, the radial wear is far less than 35 mm, and it can still be used normally. In contrast, in reference to wear failure cutter, after uniform wear and tear, the radial wear basically reaches 35 mm, which cannot be used normally. Angled wear failure is an abnormal wear failure caused by continuous wear in a local area of the cutter ring when the cutter is stuck and cannot rotate. The rock samples used in the test are granite (typical hard rock) and limestone (typical soft rock), as shown in Appendix A. Some physical parameters of the two rock samples are shown in Table 2.

### 2.2. Cutter Test Bench

The test uses the tool linear cutting test bench provided by the cooperative unit, which can simulate the cutting process of the disc cutter on the rock under the conditions of different penetration and cutting speed. In reference to penetration, i.e., the depth of the cutter pressed into the rock sample when breaking the rock, its value is controlled by the spiral screw device of the test bench. The cutting speed is controlled by the axial shift cylinder of the test bench. The specific structure of the test bench is shown in Figure 3 and Appendix A, and the specifications are shown in Appendix A. During the test, the rock sample was fixed in the rock box on the foundation cap by tightening the bolts to prevent the rock from being displaced during the simulated rock breaking process. We used bolts to fix the knife holder on the movable platform. The screw adjusts the vertical position of the cutter. The axially moving oil cylinder controls the movement of the rock box to simulate the cutting process during rock breaking.

### 2.3. Vibration Signal Collection

#### 2.3.1. Sensor Selection

Compared with other types of sensors, piezoelectric sensors have the characteristics of large dynamic range, wide frequency range, simple structure, robustness and durability, and less external interference, and are widely used in vibration measurement. The three-axis acceleration sensor has significant advantages on occasions where the direction of movement of the object cannot be determined in advance. It can comprehensively and accurately measure the spatial acceleration of the object. Therefore, in order to more accurately and comprehensively understand the vibration characteristics of the disc cutter during rock breaking, an IEPE (Integrated Electronics Piezo-Electric) three-axis acceleration sensor was used to collect the vibration signal of the vibrating cutter during rock breaking. Specifically, the 3263A3 three-axis sensor (Dytran Instruments, Inc., Los Angeles, CA, USA) was used. Under complex environmental conditions, this type of sensor is sturdy and durable, has strong anti-interference ability, good sealing performance and high reliability. The shape and parameters of the sensor are shown in Figure 3.

#### 2.3.2. Sensor Layout

Considering that different parts of the cutter may have different vibration characteristics, three vibration-signal collection positions were selected in this experiment, namely the cutter ring, the cutter hub and the cutter seat. The three specific positions of the sensor are shown in Figure 3 and Appendix A. The signal-collection position on the cutter ring is close to the contact area between the cutter and the rock sample. The vibration signal is obvious and the vibration characteristics are abundant. Therefore, this paper mainly focuses on the feature extraction and analysis of the vibration signal at this position. In particular, Figure 3 shows the relative spatial positions of the X, Y, and Z axes when the three-axis sensor is placed at the position of the cutter ring. The detection results of the three axes of the sensor show that the X vibration signal is the most obvious and this feature is the strongest. Therefore, this paper gives priority to the analysis of characteristics in the time domain and frequency domain based on the *X*-axis vibration detection results at the position of the cutter ring.

#### 2.3.3. Signal Processing and Transmission

The piezoelectric signal collected by the sensor is transmitted to the SIRIUS data acquisition module (DEWEsoft, Inc., Trbovlje, Slovenia) through the data line for processing, and the analog quantity is converted into a digital quantity, and then the processing result is transmitted to the PC through the USB data line to the DEWEsoft X3 12 software platform for storage. The FFT analyzer module provided by X3 has most of the functions for spectrum analysis, which is convenient for frequency domain analysis of vibration signals.

### 2.4. Rock Breaking Test

The axial moving speed of the rock box of the selected test bed was 30 mm/s. The penetration parameter of the cutter rock-breaking test was 2 mm. Before the rock-breaking test, the cutters in each wear state were subjected to an empty stroke test, and vibration signals were collected at three signal collection positions. The analysis results show that the vibration generated by the operation of the test bench is very weak, and there is no characteristic in the frequency domain, which basically has no effect on the test results. The sampling rate of the SIRIUS module is 5000 Hz, the FFT analysis uses the Blackman limit window, the resolution is 1 Hz, the frequency domain range is 0~2500 Hz, and the output time interval is 1 s.

## 3. Results

In Section 3.1, the time-domain and the frequency-domain analyses are performed on the vibration signal during cutter-granite contact for cutters with different wear status. The granite is a type of typical hard rock that TBM often encounters. The analysis results show that the vibration characteristics can be used to identify the wear status of the cutter during cutter-granite contact. However, it is not known whether the vibration signal still retains the characteristics related to cutter wear, when changing the working conditions and geological conditions. In Section 3.2, Section 3.3, Section 3.4, the effects of geological conditions and working conditions on vibration characteristics related to wear status are discussed. These findings are also helpful to understand the rock damage mechanism and determine the application scope of the wear status identification method based on vibration signal.

### 3.1. Analysis of Rock-iBreaking Vbration on Granite for Cutters with Different Wear Status

#### 3.1.1. Time-Domain Analysis

For the vibration signal generated when the cutter breaks the rock on granite with a penetration of 2 mm, the detection result for cutters with different wear status is shown in Figure 4. One stroke of the cutter is divided into two parts. After the positive stroke is completed, the cutter completes the return stroke along the positive stroke path in the opposite direction. In each stroke, the actual moving speed of the rock box fluctuates within a certain range, and the contact time between the cutter and the rock is about 50 s, so the time domain waveform of 60 s (including the complete positive stroke) is taken for analysis. It can be seen from Figure 4 that there are great differences in the vibration waveforms in different time areas. According to whether the vibration has periodic characteristics, we artificially divide it into two areas A and B. A1, A2, and A3, respectively, represent the generated periodic waveform when the normal wear cutter, the wear failure cutter and the no-angled wear area of the angled wear failure cutter breaks the rock. B1, B2, and B3 represent corresponding non-periodic vibrations. Different cutters and different rock breaking conditions will produce different local vibration waveform characteristics during a rock-breaking stroke. The typical waveform characteristics are selected for specific analysis and obtained in Figure 5.

In the areas of A1 (corresponding to the normal wear cutter), A2 (corresponding to the wear failure cutter), and A3 (corresponding to the angled wear failure cutter), the surface of the rock sample has powdery fragmentation, and the vibration waveform has obvious periodic characteristics, as shown in Figure 5a (wherein, red, blue, and yellow represent the periodic waveforms in the A1, A2, and A3 regions, respectively). From the point of view of system dynamics, the system with friction is a nonlinear dissipative open system, that is, the system may generate periodic vibration when subjected to micro-perturbation [35]. Due to the discontinuity of the actual contact between the cutter and rock, the real friction process at the microscopic level is continuously unstable. However, when powdery fragmentation occurs at the early stage of rock-breaking, the actual contact stress between the cutter-rock is small, and the load changes little, resulting in micro-oscillation on the contact surface of the cutter-rock and corresponding periodic vibration.

In the areas of B1 (corresponding to the normal wear cutter) and B3 (corresponding to the angled wear failure cutter), the rock sample produces deep and long cracks, which are called main cracks. The main crack grows downward, resulting in a sudden change in the rock-breaking load with violent vibration and impact. Because of the main crack, the contact between the cutter and rock is unstable. Therefore, a single zigzag wave crest appears in the time domain (Figure 5b, where red and yellow represent the waveform in the B1 and B3 regions, respectively). In the area of B2 (corresponding to the wear failure cutter), the surface of the rock sample is mainly broken in block forms, so the vibration wave peaks suddenly change and then gradually attenuate (Figure 5b, where blue represents the waveform in the B2 area). In addition, the detection results in the *Y*-axis and *Z*-axis directions of the sensor also have periodic waveform characteristics in the time domain, as shown in Appendix A.

#### 3.1.2. Frequency-Domain Analysis

Comparing the periodic waveform area A1 with the non-periodic area B1 for normal wear cutter, it is found that the vibration and shock accompanying the appearance of deep and long cracks will cause component concentration in the frequency domain (Figure 5d), and the acceleration component concentration will cause the frequency domain characteristics of the nearby area to be masked. After eliminating the effects of deep and long cracks, it can be seen that there are obvious features in the rock-breaking vibration signal of the cutter (Figure 5c). 

Figure 6a shows the frequency-domain characteristics of the 0~400 Hz frequency range in the A1 area for normal wear cutter. In the range of 0~100 Hz, it can be seen that the acceleration components of the stable periodic wave are mainly distributed in six frequency bands (about 12 Hz~18 Hz, 28 Hz~33 Hz, 45 Hz~48 Hz, 58 Hz~63 Hz, 73 Hz~78 Hz, 87 Hz~95 Hz). Figure 6b shows the 0~400 Hz frequency-domain characteristics of the A2 area for wear failure cutter. Similarly, in the range of 0~100 Hz, the acceleration components are mainly distributed in six frequency bands (about 15 Hz~21 Hz, 30 Hz~35 Hz, 48 Hz~52 Hz, 65 Hz~70 Hz, 82 Hz~85 Hz, 99 Hz~104 Hz). Through comparison, it can be concluded that in the range of 0~100 Hz, when the characteristics of stable periodic waves appear in the time domain, the frequency domain shows the characteristics of vibration acceleration components mainly distributed in six frequency bands; compared with the normal wear state, the six frequency bands corresponding to the wear failure state all move to a higher frequency direction. Comparing the frequency-domain characteristics of the 100~400 Hz frequency range (Figure 6a,b), it can be seen that the normal wear status has almost no acceleration component, while the wear failure status has more acceleration components and is distributed in multiple frequency bands. 

Figure 6c shows the vibration-detection results in the process of rock breaking in the non-angled wear area for angled wear failure cutter. In the non-angled wear area, the cutter wear state is basically the same as the wear failure state of the cutter, so the frequency-band characteristics also appear in the frequency domain. Figure 6d shows the vibration-detection results in the process of rock breaking in the angled wear area for angled wear failure cutter. It can be seen that the cutter will vibrate violently in the angled wear area. At the same time, the prominent feature in the frequency domain is that a large number of high-order harmonics are excited.

### 3.2. Analysis of Rock-Breaking Vibration with Different Rock Conditions

#### 3.2.1. Analysis of Rock-Breaking Vibration on Limestone

Figure 7a shows the typical result of the rock-breaking test on limestone for the normal wear cutter. Because the vibration characteristics of the wear failure cutter are similar, it will not be described in detail here. The surface of the limestone sample is mainly fragmented in block form (Figure 8a), so there is no stable periodic vibration waveform, and most of it is a characteristic waveform that suddenly changes and then gradually attenuates, as shown in Figure 7b. Compared with granite, the hardness of limestone is much lower, and the vibration generated when the same cutter breaks the soft rock is relatively gentle. It shows strong randomness in the frequency domain, and the abrupt changes are often unstable.

Figure 7c,d shows the vibration detection results in the process of rock breaking for the angled wear area of angled wear failure cutter. Due to the geometric defects of the friction pair surface, that is, the uneven flatness of the angled wear surface, the cutter has discontinuous rotation and sliding. This repeated cyclic stick-slip motion can cause strong and long-lasting vibrations [36]. This process is accompanied by frictional heating and uneven distribution of the friction system caused by the surface of the rock sample being crushed by friction, which intensifies the uneven wear of the angled wear surface. This process may be unstable. The vibration characteristics of the angled wear area when the rock is granite is shown as Figure 6d, which also reflect the characteristics of strong vibration with high-order harmonics. When the rock is limestone with softer texture, the repeated cyclic stick-slip motion also causes periodic vibration, but the generated vibration is relatively gentle and does not excite high-order harmonics. The acceleration components are mainly distributed in multiple frequency bands from 0 to 600 Hz in the low frequency range (Figure 7d). It can be concluded intuitively that even if the same cutter and working condition are used, the vibration signal becomes different due to different rock characteristics. Due to the unstable and random vibration, it is quite difficult to effectively distinguish different cutter wear states under typical soft rock such as limestone.

#### 3.2.2. Analysis of Vibration Distribution Mechanism 

To explain the vibration-distribution mechanism from the theoretical perspective to support the experimental work, we supplement the correlation between the rock cleavage mode and the vibration distribution, which can be divided into the following questions:How does periodic vibration occur?

From the point of view of system dynamics, the system with friction is a nonlinear dissipative open system, that is, the system may generate periodic vibration when subjected to micro-perturbation. Due to the discontinuity of the actual contact between the cutter and rock, the real friction process at the microscopic level is continuously unstable. However, when powdery fragmentation (compact core area) occurs, the actual contact stress of the cutter-rock is small, and the load changes little, resulting in micro-oscillation on the contact surface of the cutter-rock and periodic vibration.

b.The relationship between the vibration characteristics of the cutter and the wear statue of the cutter (the cutter profile changes from circular-tip to flat-tip, as shown in Figure 8b)?

As shown in Figure 8a, when breaking rock, firstly, a compact core area is formed in the rock directly below the cutter (powder fragments are generated). As the load further increases, the cracks initiate, expand, branch, and merge outside the compacted core until the rock collapses and breaks (the main crack grows downward, resulting in a sudden change in the rock breaking load, so a single zigzag waveform appears; laterally growing cracks lead to fragmentation in block form, unstable cutter-rock contact, and also make periodic vibration disappear). The differences in rock fragmentation are attributed to changes in the stress field within the rock, which are strongly influenced by the cutter profile. The normal wear cutter (circular-tip cutter profile) is in line contact with the rock sample, and the wear failure cutter (flat-tip cutter profile) is in surface contact with the rock sample. Therefore, under the same normal load, the contact area of the circular-tip cutter profile is smaller than that of the flat-tip cutter profile, and the stress field is more concentrated, resulting in higher contact pressure and higher corresponding internal stress, which is more conducive to crack initiation. Therefore, under the same rock-breaking load conditions, the more severe the wear of the cutter, the flatter the surface of the cutter profile, the easier it is to cause powdery fragmentation, and the more obvious periodic vibration characteristics. 

c.Factors affecting periodic vibration characteristics

① The wear of cutter

With the increase in the wear of the cutter, it appears to migrate to high frequency in the frequency domain, and the high frequency component increases. We believe that there is a correlation between the harmonic components of the vibration signal and the natural frequencies of each order of the cutter-rock system. The cutter is composed of several components and is subject to friction during rock breaking. The friction surface of normal wear cutter (circular-tip cutter profile) is small, so the amplitude of natural frequency is high and the amplitude of harmonics is generally not too high. However, as the wear increases, the friction area increases, and the friction acts as an additional support. The amplitude of the natural frequency decreases and the vibration frequency increases, while the amplitude of the harmonics increases due to the additional linear effect. If the friction area is further increased (such as the angled wear failure cutter), a strong frictional tangential force will be generated, which can completely destabilize the rotating system of the cutter. At this time, the vibration response has a high sub-harmonic component. 

② Geological conditions

In the rock breaking process, the cutting force on the rock is mainly divided into three types, namely edge lateral force, normal thrust and rolling force. The lateral force on the rock is much smaller than the other two forces, so it can be ignored. Therefore, the rock breaking is mainly realized under the combined action of the normal thrust and the rolling force [37]. Under the combined action of the two forces, the rock first undergoes elastic deformation, then plastic deformation, and finally yields and breaks. The yielding and crushing stage of rock is composed of the generation, expansion and intersection of rock cracks [36]. Rock hardness and strength are important factors that affect the formation and propagation of cracks. In addition, the strength of the rock joints that may exist is different from the strength of the rock itself. This difference has a greater impact on the cracks and extension directions of the rock under the force of the cutter [36]. In our experiment, Hard rock (granite) is dominated by elastic brittle failure, and the strain energy accumulates in the rock (the compact core area is formed first) to a certain extent and is suddenly released (crack growth). Therefore, the damage mechanism of hard rock determines that rock breaking by the cutter must be a dynamic process accompanied by vibration and impact. When the cutter is in stable contact with the rock and the rock presents powdery fragmentation, the vibration signal has periodic characteristics.

The soft rock is dominated by plastic failure. With the large deformation of the rock, the vibration displacement of the cutter will be affected by the rock damping. At this time, most of the impact energy of the cutter on the rock is consumed by the damping, so there is almost no periodic vibration, but a gradually decaying vibration signal is replaced, which is consistent with the experimental phenomenon. According to observations, granite samples mainly have two types of fragmentation: powdery and deep and long cracks, while limestone samples are mainly fragmented in block form. Therefore, the current method of identifying and diagnosing the wear status of cutters based on the vibration frequency domain characteristics used in this experiment is suitable for hard rock formations represented by granite.

③ In addition, according to the above analysis, the penetration degree, the confining pressure and the spacing of the cutter may have an impact on the vibration of the cutter [38]. Section 3.4 discusses the effect of the penetration degree on the vibration distribution. 

### 3.3. The Influence of Different Sensor Positions

Among the three sensor detection positions selected in this experiment, the position of the cutter ring and the cutter hub is closest to the rock contact area of the cutter, and a wealth of vibration characteristics can be collected. However, in practical applications, the sensor system needs to be embedded in the disc-cutter body, which will increase the complexity of the cutter design, and these locations are the most severely impacted by muddy water, and the environment is the worst. Compared with the cutter ring and the cutter hub, the position of the cutter seat is easier to be structurally modified to install the sensing system, which has stronger engineering practical value. Therefore, we measure the rock-breaking vibration signals at different sensing positions. Appendix A shows the vibration detection results of the sensor at the three positions of the cutter ring, cutter seat and cutter hub when the cutter is used for the rock-breaking test on the granite sample. It can be seen that the periodic vibration waveform can be detected during a certain period of time at the position of the cutter seat and cutter hub. Since the sensor can accurately perceive the periodic waveform, it is possible to place the sensor on the cutter seat or the cutter hub to diagnose cutter wear status by detecting the vibration signal. Its feasibility needs to be further verified by subsequent in-depth tests in the future. 

### 3.4. The Influence of Different Penetration 

Penetration refers to the depth to which the cutter is pressed into the rock sample when breaking the rock. In addition to the cutter wear status and rock characteristics, penetration is also an important working condition parameter that affects vibration characteristics. Appendix A, respectively, shows the vibration-detection results of the normal wear cutter at the position of the cutter ring under the working conditions of penetration of 1 mm, 2 mm, 3 mm, and 4 mm. The rock-breaking load does not increase uniformly with the increase in the penetration degree, but presents the characteristics of leap and sudden change. When the penetration is small (~1 mm), the contact between the circular-tip cutter profile of the normal wear cutter and the rock is unstable and the contact area is small, resulting in insufficient contact between the cutter and rock, and the compact core area cannot be effectively formed in the rock, with no occurrence of periodic vibration. On the contrary, the flat-tip cutter profile of the wear failure cutter is in stable contact with the rock and has a large contact area, resulting in emerging compact core area and periodic vibration for penetration of 1 mm (Appendix A). This phenomenon also supports the previous point of view, that is, the greater the degree of wear of the cutter, the flatter the surface of the cutter tip and the easier it is to cause powdery fragmentation, and the more obvious are the periodic vibration characteristics. When the penetration is large (≥4 mm), the contact stress of the cutter-rock increases, resulting in the initiation of cracks and the disappearance of periodic vibration. When the penetration is moderate (2~3 mm), both the circular-tip and flat-tip cutter profile have sufficient contact area with the rock, and powdery fragments appear, generating periodic vibrations. Comparing the four penetration working conditions, it can be seen that the method of judging the wear status of the cutter by the vibration signal can be applied more stably within a certain range of penetration, such as 2 mm and 3 mm. 

### 3.5. Test to Identify Wear Status Based on ANN

According to the aforementioned research, considering the stability and reproducibility of the vibration signal, we install the sensor on the cutter ring and collect vibration-time domain signals generated by the single-edged disc cutter breaking the rock on the granite samples. We use the FFT method to convert the time-domain signals into the frequency-domain signals and output them as time-series datasets. Pytorch is used to build a Long- and Short-Term Memory network model (LSTM), and the output datasets are used to train the model, so that the model can identify and extract frequency domain features and establish a correlation with the wear status of the cutter. The test results show that the accuracy of the model can reach more than 90%.

The vibration signal in the rock-breaking process of the cutter is complex and changeable, and the characteristic signal is often hidden in a large number of noise interference signals. It is difficult for conventional signal processing methods to efficiently identify vibration-signal characteristics in a large number of noise signals according to preset standards, and it is even more difficult to extract characteristic signals with general regularity. Deep learning has the potential to objectively learn representative features of raw data. The LSTM model has advantages in solving sequence problems with complex temporal correlation and long-term memory of historical information. Therefore, this method uses the LSTM model, with its excellent feature extraction ability and its ability to process time-series signals, to extract vibration characteristic signals with general regularity from a large number of input signals for cutter wear identification. The adaptive ability of the model can get rid of the dependence of artificial experience and learn recognition criteria by itself. During the recognition process, the model with self-learning and strong memory ability can continuously strengthen and update the extracted vibration characteristic signal, and improve the recognition efficiency and accuracy.

#### 3.5.1. Model Framework

The Recurrent Neural Network (RNN) takes sequence data as input; all nodes are connected in a chain, and it has short-term memory capabilities. It has certain advantages when learning the nonlinear characteristics of sequence data. In RNN, “neurons” can receive information from themselves and other “neurons” to form a network structure with loops. However, RNN has an obvious problem, Long-Term Dependencies, that is, the state a long time ago needs to affect the current state, but too large an interval between the two may reduce this influence. LSTM is one of the common forms of RNN. It was first proposed in 1997 and was designed for the long-term dependence of ordinary RNN. An outstanding feature of LSTM is the introduction of gating mechanisms (forgetting gates, input gates, output gates), which are suitable for processing and predicting events with very long intervals and delays in time series, and can be used as non-linear units to construct large-scale deep neural networks.

#### 3.5.2. Dataset Construction

It can be seen from the foregoing that when the disc cutter is used in the rock-breaking test on the granite sample, the vibration characteristics are relatively rich, and it has the conditions for identifying and diagnosing the different wear conditions of the cutter. Therefore, this test uses the vibration-frequency domain characteristics of the granite rock-breaking test to construct a time-series dataset. The sensor is used to collect the vibration acceleration signals of the cutter in different status on the granite sample with the penetration of 1 mm, 2 mm, and 3 mm in the rock-breaking test at the position of the cutter ring, and convert them into piezoelectric signals. The data acquisition module samples the piezoelectric signal at a sampling rate of 5000 Hz to obtain the vibration-time domain signal, which is transmitted to the data processing output module. The data-processing output module uses the FFT method to convert the time-domain signal into the frequency domain signal with a resolution of 1 Hz, and the frequency-domain range is 0~2500 Hz. We visually output the time-frequency domain data to the PC, and selected signal segments with obvious time-frequency domain characteristics. The selected signal segments are output as time-series frequency domain data at 1 s intervals to form a dataset, which is divided into training, validation, and test datasets according to the ratio of 7:2:1. In each dataset, the wear status of the cutter corresponding to each piece of data is called a label.

#### 3.5.3. Model Training and Optimization

This test trains an LSTM network model based on the correspondence between the vibration frequency domain characteristics and different wear status (normal wear status, wear failure status, and angled wear failure status) of the cutter (Figure 9a). This experiment uses the Pytorch framework to build an initial LSTM model with a one-way, two-layer, hidden size of 128 in the Pycharm integrated environment. The training optimizer is Adam; the learning rate is 0.008, and the CrossEntropyLoss function is used to calculate the loss value. After completing the parameter initialization, we imported the initial model to the GPU device to achieve fast training.

In the training process, the training dataset is cyclically imported into the model, and each import is recorded as a epoch. In each epoch, the training data set is imported into the model in batches for training, and the batch size is 128. Input data and model parameters are subjected to mathematical operations to obtain output; output and label are imported into the loss function to obtain loss; the gradient is calculated from loss, and the model parameters are updated through backpropagation. At the same time, perform a Boolean operation on output and label to get the training accuracy (denoted as “Train_acc”). The “Train_acc” and loss of the whole training process are shown in Figure 9b. In addition, the learning rate decay function is added. Every two training epochs, the learning rate becomes 95% of the previous one, avoiding that the gradient descent cannot converge to the global optimum as the training epoch increases.

The neural network model continuously updates the parameters, so that the output gradually fits the label, reduces the loss and improves the accuracy. In fact, there is “interference” in the training data. If the model remembers the features of “interference” too much, the accuracy of the training dataset (“Train_acc”) will be significantly improved. However, the accuracy of the model on other datasets will be reduced, that is, the generalization ability of the model will be reduced. This phenomenon is called Over-Fitting. In order to prevent Over-Fitting, the model needs to be optimized to reduce the risk. Firstly, we increased the Early-stopping function. Whenever the model completes an epoch of learning, we imported the validation dataset to obtain the validation accuracy (“Valid_acc” in Figure 9b). If the “Valid_acc” continues to decrease in 24 epochs, we stop the training process and save the model parameters corresponding to the optimal “Valid_acc”. Secondly, we increased the Dropout function. During each training, a part of “neurons” are randomly selected and set to 0 (inactivated) according to a certain probability, so that they will not participate in this training. This function can reduce the coupling between “neurons” and force each “neuron” to extract appropriate features to increase the generalization ability of the model.

After the model training is completed, we imported the test dataset into the model to obtain the test accuracy (“Test_acc” in Figure 9b). It can be seen from Figure 9b that the model does not appear to be Over-Fitting, and the test accuracy rate (“Test_acc”) exceeds 90%. Figure 9c shows the recognition results of the test dataset in matrix form. Among them, True label represents the true wear status of the cutter, and Pre result represents the status of the cutter identified by the model. The three pieces of data on the diagonal indicate the number of correct recognition, which is far more than the number of incorrect recognition. As can be seen from the training results, the model exhibits an accuracy rate (Figure 9b) of over 90%. Therefore, it is feasible to use the LSTM neural network model to identify and diagnose the tool wear state based on the vibration-frequency domain features.

## 4. Discussion

However, the current limitation is that the generation of vibration characteristics requires that the rock-breaking load of the cutter does not undergo a step mutation, and the rock surface can stably form a compact core area when the rock fragments are powdery. This paper verifies a typical scenario that satisfies the conditions, that is, periodic vibration occurs when the corresponding rock powder is generated in the early stage of elastic brittle failure of hard rock granite. Some parameters affecting the rock cleavage, such as cutter wear, geological conditions, penetration degree, and even cutter layout (spacing between two cutters), rock confining pressure, etc., will theoretically change the rock-breaking efficiency and rock cleavage form, resulting in the migration or even disappearance of vibration characteristics. That is, in the actual application process, the migration of vibration features can be solved by training a neural network model with a large amount of data, and the disappearance of vibration features (soft rock, or excessive penetration) might lead to the failure of this method.

It is worth noting that, for hard rock, this method seems effective. Additionally, in reference to some abnormal wear status, such as angled wear failure state, whether for hard rock or soft rock, the monitoring method in this paper shows extremely typical vibration high-order harmonic characteristics, which has potential significance for the monitoring of cutter abnormal wear state in engineering practice. Although this work is still far from practical engineering applications, which require a large amount of engineering experimental data to establish engineering databases and models, the new monitoring method proposed in this paper provides a theoretical basis for later engineering practice.

## 5. Conclusions

This work proposes a new monitoring method to identify the wear status of a single TBM single-edged disc cutter based on vibration signals and machine learning. The main conclusions are as follows:

① We discussed and analyzed the distribution mechanism of the vibration signal, and found that the distribution of the vibration signal is highly correlated with the rock-cleavage mode; the rock-cleavage mode is directly modulated by the cutter wear status, thus obtaining the corresponding relationship between the vibration signal and the cutter wear.

② In the actual rock-breaking process, complex working conditions and geological conditions may be faced. We selected different geological and penetration parameters to discuss the feasibility and limitations of this method in practical engineering applications, which provided a reference for the subsequent engineering application of this method.

③ Although deep-learning algorithms have shown value in routine tool wear diagnosis, there are currently few studies and applications in TBM disc-cutter wear diagnosis. In this paper, an appropriate deep-learning algorithm is selected to diagnose and identify the abnormal wear state of the cutter combined with the vibration signal. With the help of the neural network model, the problem that it is difficult to accurately identify the vibration characteristics of the cutter under noise interference is solved, and the cutter with different wear status during cutter-hard rock contact could be identified with a diagnosis accuracy rate of 90%.

④ The periodic vibration signals could be detected at multiple locations, such as at the cutter shafts, which is conducive to the safe and firm installation of the sensor and avoids the interference of complex environments to a large extent.

## Figures and Tables

**Figure 1 sensors-22-06686-f001:**
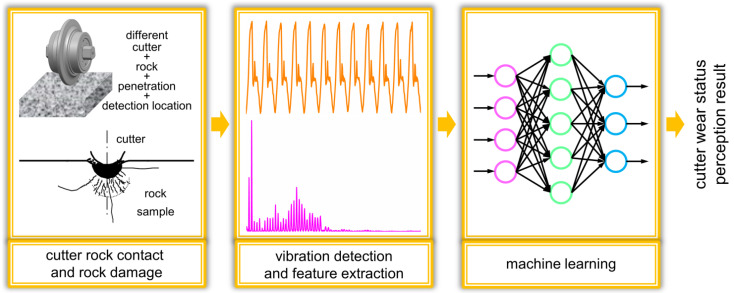
Cutter wear identification process. The lines in the middle figure show the vibration characteristics in the time and frequency domains; the right figure shows the neural network model algorithm.

**Figure 2 sensors-22-06686-f002:**
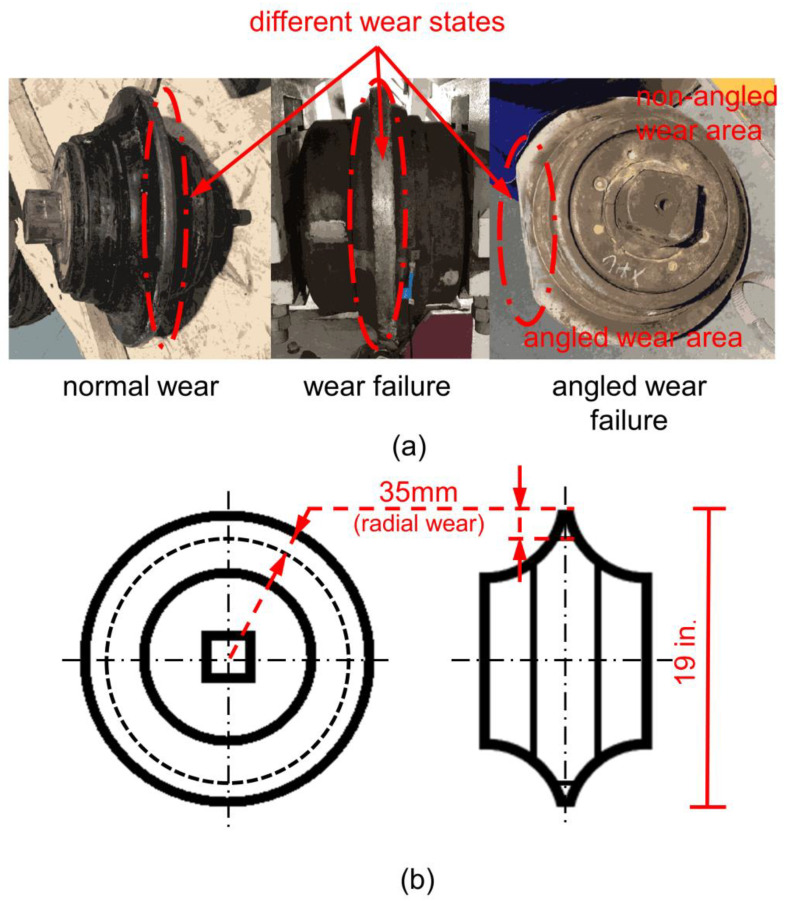
19 in. single-edged disc cutter used in the test: (**a**) 3 types of single-edged disc cutters; (**b**) judgment criteria for cutter wear failure, radial limit wear 35 mm.

**Figure 3 sensors-22-06686-f003:**
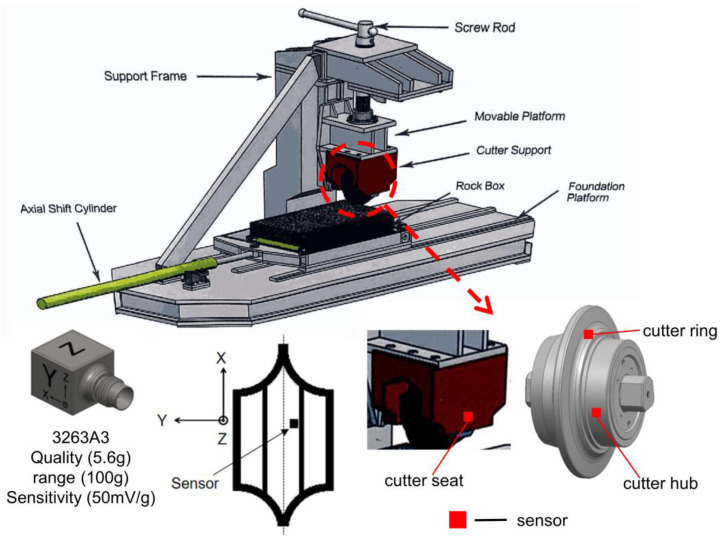
Rock-breaking cutter test bench, and corresponding sensor parameters and layout.

**Figure 4 sensors-22-06686-f004:**
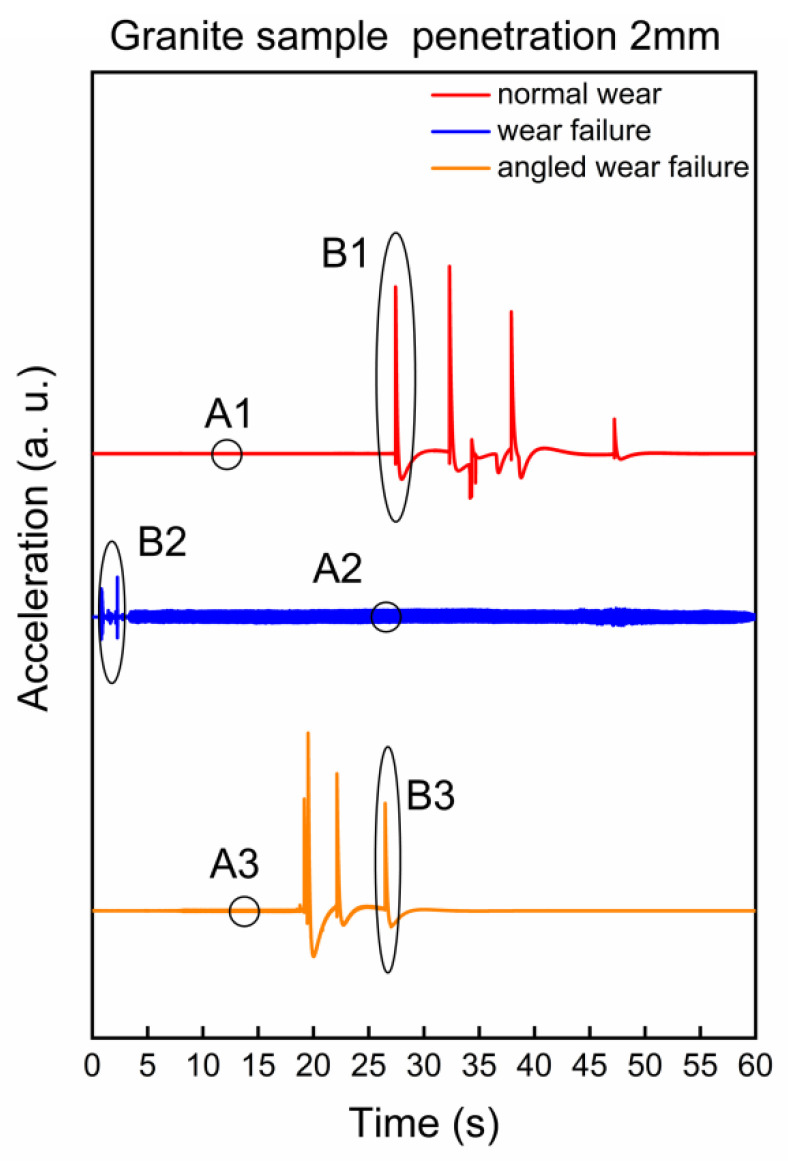
Granite sample, penetration 2 mm, 60 s time domain vibration waveforms of rock breaking process with different cutters.

**Figure 5 sensors-22-06686-f005:**
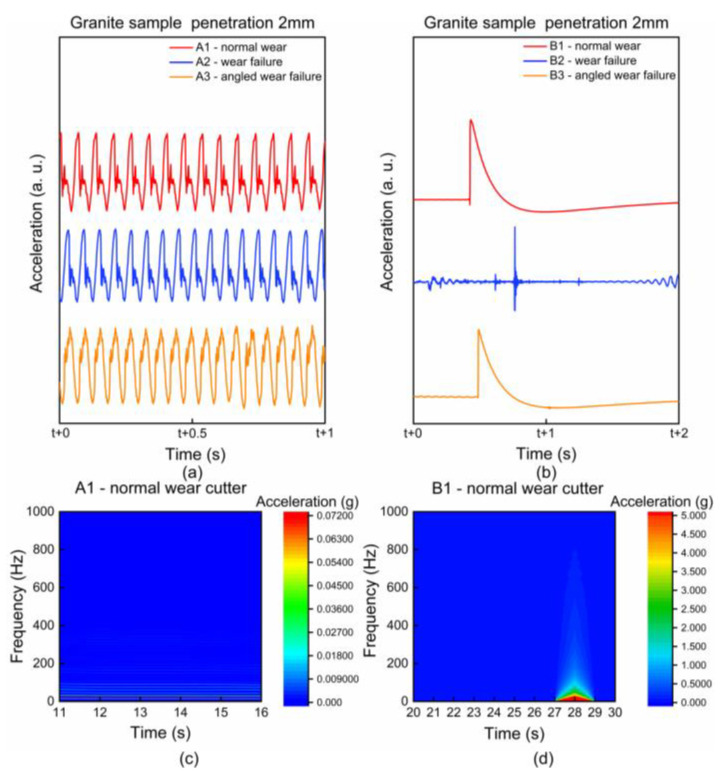
Granite sample, penetration 2 mm, comparison of periodic and non-periodic regions of different cutters in the time domain: (**a**) periodic waveform in area A1, A2, A3; (**b**) wave characteristics of areas B1, B2, B3; (**c**) A1 periodic waveform area, 0~1000 Hz, frequency-domain characteristics; (**d**) B1 deep- and long-crack area, 0~1000 Hz, frequency-domain characteristics.

**Figure 6 sensors-22-06686-f006:**
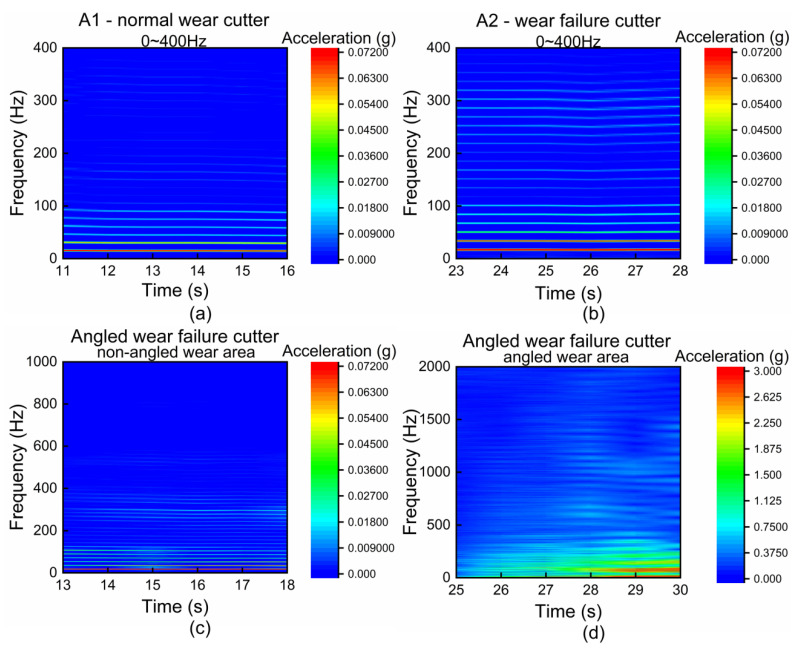
Granite sample, penetration 2 mm, frequency domain characteristics of different cutters: (**a**) normal wear cutter, A1 area, 0~400 Hz; (**b**) wear failure cutter, A2 area, 0~400 Hz; (**c**) angled wear failure cutter (non-angled wear area), 0~1000 Hz; (**d**) angled wear failure cutter (angled wear area), 0~2000 Hz.

**Figure 7 sensors-22-06686-f007:**
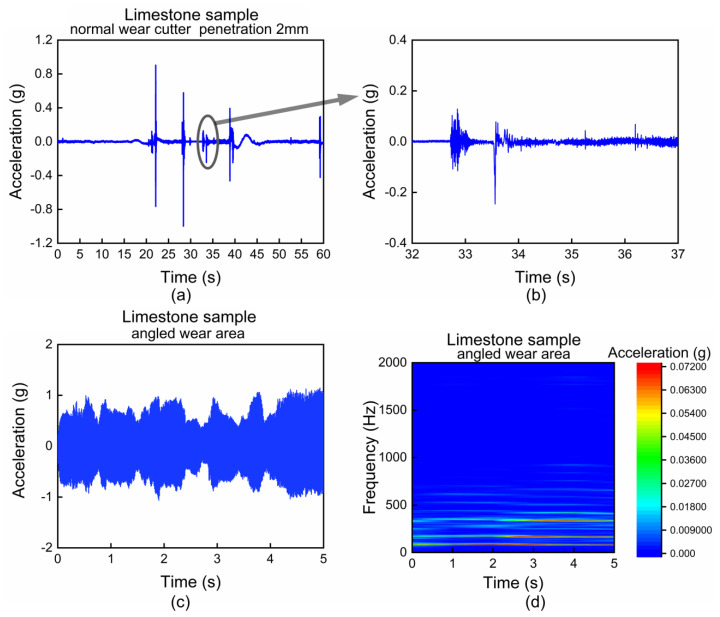
Limestone sample, vibration characteristics: (**a**) normal wear cutter, penetration 2 mm, 60 s time domain waveform; (**b**) normal wear cutter, the waveform in selected area, no periodic characteristics; (**c**) angled wear failure cutter, angled wear area, time domain waveform; (**d**) angled wear failure cutter, angled wear area, frequency–domain characteristics.

**Figure 8 sensors-22-06686-f008:**
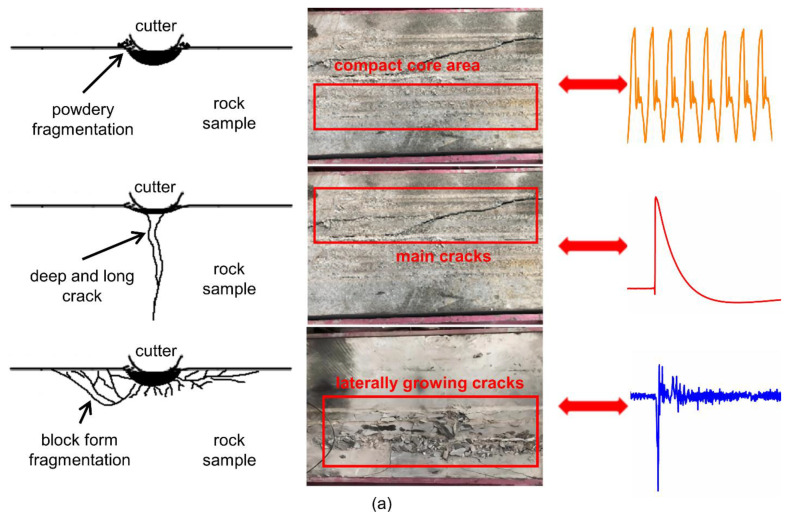
Rock Fragmentation and cutter profiles: (**a**) three main cleavage types of rock samples; The three colored lines indicate the vibration time-domain characteristics of the corresponding rock states, respec-tively; (**b**) flat-tip cutter profile and circular-tip cutter profile.

**Figure 9 sensors-22-06686-f009:**
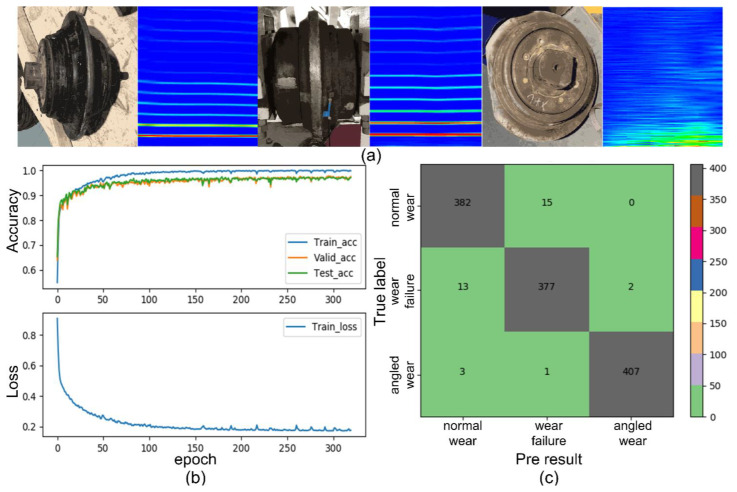
Neural network model recognizes cutter wear status based on frequency domain characteristics: (**a**) correspondence between vibration characteristics and wear status; (**b**) model accuracy and loss; (**c**) comparison of forecast results. Each background color corresponds to a numerical range, and the background color indicates the number of occur-rences of each situation.

**Table 1 sensors-22-06686-t001:** Comparison between different wear identification techniques.

Method	Feature	Advantage	Shortage
Hydraulic or smell inspection	Renovate the inside of the cutter, add hydraulic pipes or odor additives	■ The judgment of the preset wear amount can be realized	♦ The hydraulic pipe leaks easily ♦ Less reliability and immunity to interference ♦ Unable to monitor online
Eddy current testing [18,19,20]	As the cutter wears, the distance between the cutter and the sensor gradually increases, causing the sensor output voltage to change	■ High measurement accuracy ■ Good anti-interference	♦ A small linear measurement range ♦ Limited-service life in slag and muddy water ♦ High installation accuracy requirements
Laser detection [22]	Obtain the real-time wear amount of the cutter according to the on-off of the parallel light path	■ Relatively simple installation	♦ Poor anti-interference ability ♦ Low reliability
Ultrasonic testing [21]	The real-time wear amount of the cutter is converted according to the time difference between the transmission and the reflected reception	■ Relatively simple installation	♦ Poor anti-interference ability ♦ Low reliability
Vibration sensor detection (Our work) [23,24]	Import the frequency-domain signal of vibration into the neural network model to complete the wear status identification	■ Easy to install ■ Good reliability and longevity ■ Good anti-interference ability	♦ Vibration characteristics may disappear in some working and geological conditions

**Table 2 sensors-22-06686-t002:** Physical parameters of rock samples.

Physical Parameter	Granite	Limestone
Density (g/cm^3^)	2.4~3.1	2.6~2.8
Elastic Modulus (kgf/cm^2^)	1.3~1.5 × 10^6^	0.2~0.8 × 10^6^
Mohs Hardness	6~7	≤3.0
Shore Hardness, HS (AVG)	78	44
Specification (mm)	1130 × 580 × 280	

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
