# Peer review of "A New Strategy for Disc Cutter Wear Status Perception Using Vibration Detection and Machine Learning"

_sensors, 2022, doi:10.3390/s22176686_

Round 1
Reviewer 1 Report
In this work, a method to diagnose the abnormal wear state of the disc cutter by using artificial intelligence (AI) to process and analyze the vibration signal in the dynamic contact between the disc cutter and the rocks. This method is mainly aimed at realizing the diagnosis and identification of the abnormal wear state of the cutter. It is an interesting applications of vibration sensors in a disc cutter. The manuscript was written in details and somewhat in an educational way. Nevertheless, it is publishable in this journal. However, more information can be provided and typos should be corrected as follows:
How to attach the sensor to the cutter if the cutter is rotating? Please provide more detail about it.
What is name of the commercial deep learning (or AI) program the authors used in this manuscript, or it was written by authors themselves?
In order to make the AI program more clearly, please add a table of input parameters and output parameters and define the meaning of those parameters in the I/O of the program.
If the table 1 can include the cited reference for each sensor, it will be better.
On line 127: “On the basis of previous research, this paper is the first…”. Please cite the references on the previous research.
On line 160: The Table 2 caption was missed.
On line 189: Please write down the full name of IEPE.
On line 191: Is “DYTRAN” a company (manufacturer) name? Please provide the company name. Also the company name for DEWEsoft SIRIUS on line 210.
On line 270 and 271: It seems Fig. 8a cited before Fig. 5a, not in sequential. The same situation in Fig. 9c comes before Fig. 9b on line 597.
On line 499: The FFT was defined at first sight before. No more repeat.
On line 529: The LSTM was defined at first sight in text before, no more repeat.
Author Response
- How to attach the sensor to the cutter if the cutter is rotating? Please provide more detail about it.
Response: Thanks for your comments. The vibration data for this work is under experimental conditions, so we stick sensors at 4 different locations on the cutter. In actual work, the structure of the cutter and the tool box seat can be modified to facilitate the installation of the vibration signal detection unit. Given the complexity of the actual working environment of cutter, this method still needs more work exploration.
- What is name of the commercial deep learning (or Al) program the authors used in this manuscript, or it was written by authors themselves?
Response: Thanks for your question. The deep learning model used in this work is an open-source LSTM model, which uses the predefined classes and functions in the Pytorch module to build the model structure and train it.
- ln order to make the Al program more clearly, please add a table of input parameters and output parameters and define the meaning of those parameters in the l/O of the program.
Response: Thanks for your comments. This work uses the open source method to build an LSTM model, takes time series data as input features, and outputs the recognition results of the model. Because the model is established by a predefined method, we have not studied the specific parameters of the model, so we cannot provide a detailed model parameter table for the time being. Your comments provide a possible exploration direction for our further research, that is, to improve the robustness and reliability of the model by optimizing the details of the parameters.
- If the table 1 can include the cited reference for each sensor, it will be better.
Response: Thanks for your opinion. The characteristics, advantages and disadvantages of each sensor detection method shown in Table 1 are partly derived from previous literature demonstrations and partly derived from practical engineering experience. We have supplemented the sensor references in Table 1 with the reviewer comments.
On line 127: "On the basis of previous research, this paper is the first...". Please cite the references on the previous research.
On line 160: The Table 2 caption was missed.
On line 189: Please write down the full name of IEPE.
On line 191: Is “DYTRAN" a company (manufacturer) name? Please provide the company name. Also the company name for DEWEsoft SIRIUS on line 210.
On line 270 and 271: It seems Fig.8a cited before Fig. 5a, not in sequential. The same situation in Fig. 9c comes before Fig.9b on line 597.
On line 499: The FFT was defined at first sight before. No more repeat.
On line 529: The LSTM was defined at first sight in text before, no more repeat.
Response: Thank you very much for your comments. We did not complete these details in the original text, and have completed the revision in the text according to the reviewer's comments.
Reviewer 2 Report
Review report of, “A new strategy for disc _______” submitted by Pu et.al to the Journal of Sensors for determining suitability for publication
My review report is as under:
1. In this research work, vibration signals are used for monitoring the cutter wear / failure etc during tunnel operations. Use of ANN is made training the data and for detection of fault.
2. What is sensor used and how it has ensured to avoid the noise?
3. ANN is used for detecting faults. The data used for training is not usable in the paper.
4. What is the accuracy achieved?
5. How the method is reliable for failure detection?
6. What is the size of cracks generated in the rock? Does the sensor pick signal from cutter only or from rock also? Then how the failures are isolated?
7. What is the hypothesis used for the detection and identification of faults? This needs to be illustrated in the paper.
8. In my opinion, data used for training ANN is very less. The authors need to generate useful data which is sufficient to train ANN
9. How many faults are detected and isolated at a time?
Author Response
1. ln this research work, vibration signals are used for monitoring the cutter wear / failure etc. during tunnel operations. Use of ANN is made training the data and for detection of fault.
Response: Thanks for your comment.
2. What is sensor used and how it has ensured to avoid the noise?
Response: Thanks for your question. In this paper, we use a piezoelectric triaxial vibration acceleration sensor, and some parameters of the sensor are marked in Figure 3 of the main text. There is a complex and violent contact behavior between the cutter and the rock, and it is currently impossible to avoid noise signals. In this paper, measures are taken to reduce the interference of noise signals: 1. Position the sensor as close as possible to the contact position between the cutter and the rock; 2. Use a smaller penetration to reduce the intensity of the contact between the cutter and the rock and reduce the noise signal; 3. The data acquisition unit used with the sensor has a certain noise filtering function.
3. ANN is used for detecting faults. The data used for training is not usable in the paper.
Response: Thanks for your comments. In this work, sensors are used to collect vibration signals generated by rock breaking, and the FFT algorithm is used to convert the vibration signals to time-frequency to obtain frequency-domain data, that is, the component value of vibration acceleration at each Hz, which expands the data dimension. The FFT-processed frequency-domain data are output in the form of time series and form a dataset for training the model.
4. What is the accuracy achieved?
Response: Thanks for your question. The trained model demonstrated over 90% accuracy on the validation dataset. We did not directly explain the accuracy rate in the original text, and now make the following modifications in Section 4.3 according to the reviewer's comments:
As can be seen from the training results, the model exhibits an accuracy rate (Fig. 9c) of over 90%. Therefore, it is feasible to use the LSTM neural network model to identify and diagnose the tool wear state based on the vibration frequency domain features.
5. How the method is reliable for failure detection?
Response: The vibration sensor is reliable and durable, and can effectively avoid the interference of high temperature, humidity and dusty environment. The vibration sensor is used to collect complex rock breaking vibration signals, and the excellent feature extraction and self-learning ability of the neural network LSTM model for time series data is used to establish the correlation between the vibration characteristics and the wear state, so as to realize the fault detection of a single cutter. This paper verifies a typical scenario that satisfies the conditions to obtain vibration data. After training and testing, the accuracy of the model can reach the expected effect, which shows that the method has certain reliability. More work is still needed for reliable and continuous fault detection.
6. What is the size of cracks generated in the rock? Does the sensor pick signal from cutter only or from rock also? Then how the failures are isolated?
Response: Thanks for your question. 1. This work mainly conducts qualitative analysis on the different crack forms produced by the rock, but does not quantitatively analyze the specific size of the crack, so we do not count the size of the rock crack. 2. The vibration signal is only collected from different positions of the cutter, and the vibration signal on the rock is not collected. 3. There are three wear states of the disc-cutter, the main difference is the radial wear amount. The normal wear cutter can still be used normally after a certain degree of uniform wear, but the radial wear is far less than 35mm. The wear failure cutter, after uniform wear, the radial wear basically reaches 35mm, which cannot be used normally. The angled wear failure cutter is the abnormal wear caused by the continuous wear of the local area of the cutter ring when the cutter is stuck and cannot rotate.
7. What is the hypothesis used for the detection and identification of faults? This needs to be illustrated in the paper.
Response: Thanks for your comments. In this work, a specific cutter rock breaking vibration test is designed, with granite as the rock sample, the disc-cutter states are set to three (normal wear cutter, wear failure cutter and angled wear failure cutter), and the fault state is two (wear failure cutter and angled wear failure cutter). The time-series data obtained through vibration experiments were used to train the LSTM model. Based on the vibration data obtained under the experimental conditions, the feasibility of the neural network model for cutter fault diagnosis is verified. The specific situation of the vibration test is described in the second chapter of the main text.
8. In my opinion, data used for training ANN is very less. The authors need to generate useful data which is sufficient to train ANN.
Response: Thanks for your comments. In this work, sensors are installed in 4 different positions to collect the vibration signal of the cutter breaking rock. Limited by the conditions of the vibration test bench, we collected as much vibration data as possible. During the manual processing and analysis of the data, it was found that some of the data could not meet the requirements of training the neural network model, thus causing the problem of the small scale of the training data set. This work uses the existing data to verify the feasibility of using the neural network model for cutter diagnosis based on vibration data, and basically achieves the expected effect. In the future, we will continue to conduct vibration tests to supplement more and more sufficient datasets to optimize the model.
9. How many faults are detected and isolated at a time?
Response: Thanks for your question. There are three possibilities for each classification result of the LSTM model: normal wear cutter, wear failure cutter and angled wear failure cutter. Among them, there are two kinds of faults, namely the wear failure cutter and angled wear failure cutter.
Round 2
Reviewer 2 Report
Suggestions implemented hence it may be accepted for publication